# Perceived Diet Quality, Eating Behaviour, and Lifestyle Changes in a Mexican Population with Internet Access during Confinement for the COVID-19 Pandemic: ESCAN-COVID19Mx Survey

**DOI:** 10.3390/nu13124256

**Published:** 2021-11-26

**Authors:** Sophia Eugenia Martínez-Vázquez, Marena Ceballos-Rasgado, Rafael Posada-Velázquez, Claudia Hunot-Alexander, Edna Judith Nava-González, Ivonne Ramírez-Silva, Daisy Karina Aguilar-López, Gabriela Quiroz-Olguín, Beatriz López-Jara, Cristina Delgado-de-la-Cruz, Sol Huescas-Juárez, Mónica Silva, Martha Kaufer-Horwitz

**Affiliations:** 1Gastroenterology Department, Instituto Nacional de Ciencias Médicas y Nutrición Salvador Zubirán, Ciudad de México 14080, Mexico; 2School of Sport and Health Sciences, University of Central Lancashire, Preston PR1 2HE, UK; marena.ceballos@gmail.com; 3Academic Technical Committee, Red de Estudios Latinoamericanos en Administración y Negocios, San Juan del Río Querétaro 76807, Mexico; rafaelposada@rafaelposada.net; 4Institute of Human Nutrition, Centro Universitario de Ciencias de la Salud, Universidad de Guadalajara, Guadalajara Jalisco 44340, Mexico; 5Facultad de Salud Pública y Nutrición, Universidad Autónoma de Nuevo León, Monterrey, Nuevo León 64460, Mexico; edna.navag@uanl.mx; 6Maternal, Children and Adolescent Nutrition Department, Instituto Nacional de Salud Pública, Cuernavaca Morelos 62100, Mexico; ciramir@insp.mx; 7Clinical Nutrition Unit, Unit of Medical Specialties in Chronic Diseases, Tula Hidalgo 42780, Mexico; karina.uec@gmail.com; 8Clinical Nutrition Unit, Instituto Nacional de Ciencias Médicas y Nutrición Salvador Zubirán, Ciudad de México 14080, Mexico; gabriela.quirozo@incmnsz.mx; 9Clínica la Jolla Echegaray, Naucalpan Estado de México 03020, Mexico; ln.edc.beatrizlopezjara@gmail.com; 10Independent Consultant, Lerma Estado de México 52004, Mexico; dcristy59@gmail.com; 11Independent Consultant, Ciudad de México 06700, Mexico; sol.huescas@gmail.com; 12Academic UNITEC, Campus Querétaro, Querétaro 76130, Mexico; monica.silva@nutrifilia.net; 13Obesity and Eating Disorders Clinic, Department of Endocrinology and Metabolism, Instituto Nacional de Ciencias Médicas y Nutrición Salvador Zubirán, Ciudad de México 14080, Mexico

**Keywords:** COVID19, confinement, lockdown, survey, diet quality, emotional eating, lifestyle, sedentary behaviour, physical activity

## Abstract

Perceived changes in diet quality, emotional eating, physical activity, and lifestyle were evaluated in a group of Mexican adults before and during COVID-19 confinement. In this study, 8289 adults answered an online questionnaire between April and May 2020. Data about sociodemographic characteristics, self-reported weight and height, diet quality, emotional eating, physical activity, and lifestyle changes were collected. Before and after confinement, differences by sociodemographic characteristics were assessed with Wilcoxon, Anova, and linear regression analyses. Most participants were women (80%) between 18 and 38 years old (70%), with a low degree of marginalisation (82.8%) and a high educational level (84.2%); 53.1% had a normal weight and 31.4% were overweight. Half (46.8%) of the participants perceived a change in the quality of their diet. The Diet Quality Index (DQI) was higher during confinement (it improved by 3 points) in all groups, regardless of education level, marginalisation level, or place of residence (*p* < 0.001). Lifestyle changes were present among some of the participants, 6.1% stopped smoking, 12.1% stopped consuming alcohol, 53.3% sleep later, 9% became more sedentary, and increased their screen (43%) as well as sitting and lying down time (81.6%). Mexicans with Internet access staying at home during COVID-19 confinement perceived positive changes in the quality of their diet, smoking, and alcohol consumption, but negative changes in the level of physical activity and sleep quality. These results emphasise the relevance of encouraging healthy lifestyle behaviours during and after times of crisis to prevent the risk of complications due to infectious and chronic diseases.

## 1. Introduction

Studies on consumer behaviours describe that purchasing habits change in external contexts such as natural disasters or pandemics. Confinement due to COVID-19 caused changes in human behaviour, services, economy, and food systems [1,2]. Initial surveys showed that during the COVID-19 pandemic, individuals were worried about their health and had made some lifestyle changes. For example, they stopped going to the gym (50%) and stocked up on food (74%) [3]. Panic purchases and hoarding of food and essential products were registered, causing shortages of some products [1,3,4]. Stockpiled foods included bottled water, milk, rice, frozen food dough, and canned foods [1,4,5].

People living in social isolation or under a crisis situation may be more likely to initiate harmful behaviours such as smoking, excessive alcohol consumption, and overeating as a mechanism of psychological relief [6,7]. Despite the fact that the participants from these studies were adults with little social interaction due to causes other than the epidemic, it is possible that the social distancing and anxiety caused by the COVID-19 state of emergency [8] had similar or even more accentuated effects on the behaviours of the population.

The results from a telephone survey conducted by the Mexican Institute of Public Health (ENSARS-CoV-2), in which data from 1000 Mexicans were collected between 11 and 30 May 2020, showed participants had less diversity in their diet, reduced their level of physical activity, and increased sedentary behaviours and sleeping hours [9]. Furthermore, according to ENSANUT 2020 COVID-19 (Mexican National Health and Nutrition Survey 2020 COVID-19), 39.5% of households presented negative dietary changes during COVID-19 confinement. Among the households that presented negative changes in their diet, 66% reported a decrease in the consumption of meat and fish, and more than 50% reported a decrease in the consumption of vegetables and fruits [9].

Mexico has a high prevalence of overweight and obesity (75.2% in adults), which places its population at high risk of severe COVID-19 [10]. Therefore, it becomes relevant to document the effects of the pandemic on the population’s eating habits and lifestyle, as an individual’s nutritional status and poor diet quality may impact on comorbidities and subsequent complications. Thus, the aim of this study is to describe the perceived changes in the quality of diet, eating behaviour, physical activity, and lifestyle of a group of Mexican adults with Internet access who were confined during the COVID-19 pandemic.

## 2. Materials and Methods

### 2.1. Study Design and Sample

For this cross-sectional study, we structured the “Encuesta sobre hábitos de alimentación y estilo de vida alrededor de la pandemia de Coronavirus (COVID-19) en México” (survey on eating habits and lifestyle around the Coronavirus (COVID-19) pandemic in Mexico: ESCAN-COVID19Mx survey) (Appendix A). Selection criteria for answering the survey were: (a) Mexican adults (aged ≥18 years), (b) any gender, (c) residing in Mexico between 17 March and 30 May 2020, (d) that confirmed staying at home during the confinement period, (e) with access to an electronic device and to the Internet, and (f) who read the informed consent form included in the online instrument and agreed to participate. The ESCAN-COVID19Mx was designed to be self-reported through an electronic platform (Google forms ©), where all responses were compulsory. The study was approved by the Ethics and Research Ethics committees of Instituto Nacional de Ciencias Médicas y Nutrición Salvador Zubirán (INCMNSZ).

The questionnaire was distributed using a snowball sampling technique through social media (WhatsApp, Telegram, Facebook, and Twitter) and email. The sample size was estimated at 2401 participants, hypothesising that a proportion of 50% of the population confined at home would present negative changes in their diet quality, with a margin of error of 2%.

### 2.2. ESCAN-COVID19Mx Survey

TheESCAN-COVID19Mx survey has five components, divided as follows: (I) sociodemographic characteristics, (II) diet quality [11] (III), emotional eating (over and under eating from negative emotions [12], (IV) physical activity and sedentary behaviour [13], and (V) lifestyle factors (quality of life). Some of the scales used as a reference in the ESCAN-COVID19Mx survey were validated in the Mexican population and could be answered in less than 15 min.

I. Sociodemographic characteristics. We collected information about the participants’ gender, age, place of residence, education, marital status, number of children in the household, and self-reported weight and height. The AMAI system 2018 [14] was used to assess participants’ socioeconomic classification, and the level of marginalisation was obtained through the participant’s postal code.

II. Diet quality. The Mini-Survey to Evaluate the Quality of Food Intake (Mini-ECCA) was included to assess diet quality during and before COVID-19 confinement. The Mini-ECCA has been validated for use in healthy Mexican adults [11]. Participants who perceived changes in their diet during the pandemic were asked to answer the Mini-ECCA twice; once in reference to their diet prior to confinement and the second time in reference to their diet during confinement. A Diet Quality Index (DQI) was calculated. Each item of the Mini-ECCA received one point if the recommendation of DQI was met. When recommendations were not met, 0 points were given. The maximum possible score was 12 points; higher values represented a better diet quality. Based on the final score, diet quality could be classified as follows: very good (10–12), good (7–9), low (4–6), and very low (1–3).

III. Emotional eating. We used the Spanish version of the Adult Eating Behaviour Questionnaire (AEBQ-Esp) validated in Mexican adults from the original AEBQ to assess emotional eating [15]. We did not assess perceived emotional eating prior to confinement, as appetitive traits have shown continuity and stability through time [16]. However, participants were asked if they thought they would have given the same answer prior to confinement. Two of the seven subscales from the AEBQ-Esp were used: (1) emotional overeating and (2) emotional undereating. Each subscale included five items each, namely: “Eat more” or “Eat less” when I am angry, worried, upset, anxious and/or irritated. Each question was answered using a five-point Likert Scale: “strongly disagree”, “disagree”, “neither agree, nor disagree”, “agree”, and “strongly agree”.

IV. Physical activity and sedentary behaviour. This component was comprised of four questions about the participant’s physical activity and sedentary behaviour during and before the contingency period. The questions about physical activity investigated the days per week that the participant performed 30 min or more of intense physical activity [13,17]. To measure sedentary behaviour, participants were asked about the hours per day they remained sitting or lying down [13,18]. We created the Physical Activity and Sedentary Behaviour Index (PASBI) with the following two scales: (a) Vigorous physical activity: participants were assigned a score (0–12) based on the times per week they completed 30 min or more of intense physical activity. These scores were distributed as follows: “did not do any vigorous physical activity” 0 points, “1–2 times per week” 3 points, “3–4 times per week” 6 points, “5–6 times per week” 9 points, and “every day or several times per day” 12 points. (b) Sedentary behaviour: Based on the time per day the participant was sitting down or lying down. The scores for sedentary behaviour were given as follows: “0 to 2 h” 12 points, “3 to 6 h” 8 points, “7 to 10 h” 4 points, and “11 h or more” 0 points. The PASBI is the result of adding both items and producing an ordinal scale of 18 levels with scores ranging from 0 to 24.

V. Lifestyle factors (Quality of life). We were interested in measuring the impact of different lifestyle factors as a composite score that comprised multiple factors [19,20]. For these questions, face and content validity were calculated among the research group and three rounds of evaluation were performed per question by consensus. Two rounds of the pilot testing were conducted in 10 and 15 individuals each, with the last approved version in the consensus included in the survey. The section was made up of the following aspects: food expenses, health information sources, compliance with confinement, changes in body weight, screen time, smoking, alcohol consumption, and sleeping habits.

#### Construction of the Lifestyle Quality Index

Information on six lifestyle factors (diet quality, physical activity, sedentary behaviour, smoking, alcohol consumption, and sleeping habits) was used to construct the Lifestyle Quality Index (LSQI). For diet, the quality response values were from 1 to 4, and for physical activity, from 0 to 4. Subsequently, these values were multiplied by 10 to weight the value of each factor equally. Responses to factors such as smoking, alcohol consumption, sedentary behaviour, and sleeping habits were first assigned a value in the range of −1 to 2, considering −1 or 0 as the least recommended or risky behaviour, and a maximum value of 2 to the recommended or non-risky behaviour. The results were divided by the total number of values of each factor, obtaining a rescaled score within each one, obtaining a maximum value of 10 points. Thus, each factor had the same weight, expressing a gradient from the lowest (0 points) to the highest (10 points) value. For the final lifestyle index, the six components were added with a maximum possible value of 60, which could be interpreted that the higher the score, the better the lifestyle quality.

### 2.3. Statistical Analyses

Sociodemographic characteristics of the participants were described using indicators of central tendency and dispersion for quantitative variables, while frequencies and percentages were used for qualitative variables. The DQI, PASBI, and LSQI were described with medians and interquartile ranges (IQR 25–75). To investigate whether there were differences in the above-mentioned indices before and during confinement, by sociodemographic characteristics and weight category, Wilcoxon’s signed ranks test and chi square were used, as appropriate. The means for each emotional eating subscale were obtained and linear regressions for both emotional over and under eating to correlate with BMI were calculated. Statistical significance was set at *p* < 0.05. All data analyses and statistical tests were performed using R (v.3.6.1, R Foundation for Statistical Computing, Vienna, Austria).

## 3. Results

### 3.1. Sociodemographic Characteristics

Respondents (*n* = 8289) fully completed the ESCAN-Covid19MX survey between 13 April and 16 May 2020. The majority of participants were women (80%), and participants were within an age range of 18 to 38 years (70%), living in areas with very low levels of marginalisation (82.8%) and had higher education levels (84.2%). Figure 1 shows the response rate by place of residence.

Half of the respondents (53.1%) had a BMI between 18 and 24.9 kg/m^2^, a third had a BMI between 25 and 29.9 kg/m^2^ (31.4%), and 13.3% had a BMI ≥ 30 kg/m^2^.

As shown in Table 1, there were significant differences in age, education, place of residence, and weight status of participants by gender. A higher proportion of women had a low and normal weight, whereas a higher proportion of men had overweight or obesity (*p* < 0.005).

The sources used by the participants to keep themselves informed about the COVID-19 pandemic were social media (78.8%); leaders and health professionals (73%); and health authorities, government, and health institutions (78.8%). Regarding money spent on food, 38.9% of the participants perceived having a higher food expenditure than usual during confinement, but 47.8% reported having a lower family income. Nearly half of the households (44%) included minors.

### 3.2. Diet Quality Index (DQI)

Nearly half of the participants (46.8%) perceived a change in the quality of their diet. There was an increase in the consumption of homemade foods (28.4%). The median of the DQI improved by three points (−1 to 2, *p* < 0.001) for the full sample. The DQI was higher during confinement in all groups, regardless of education level, marginalisation level, or place of residence (*p* < 0.001). Moreover, there was a positive association between BMI and the DQI (*p* < 0.001). Table 2 shows the changes in the DQI by age, nutritional status, schooling, degree of marginalisation, and geographic region of residence.

Figure 2 shows the change rate in the full sample in the Mini-ECCA scale before and during confinement, suggesting a shift to better diet quality.

### 3.3. Emotional Eating

Our results showed that 70% of the respondents perceived no changes in their emotional eating during confinement. The results from the AEBQ-Esp Scale showed that the emotional overeating subscale mean was 2.63 (±0.88), whereas the mean for the emotional undereating subscale was 2.51 (±0.97). Linear regressions between emotional overeating and emotional undereating with BMI showed no significant relationships between either subscales and BMI (ß = −0.937, *p* = 0.854 and ß = −1.19, *p* = 0.806, respectively).

Physical Activity and Sedentary Behaviour Index (PASBI) Participants who did not engage in any intense physical activity prior to confinement reported engaging in more intense physical activity during quarantine, as the rate of people engaging on intense physical activity increased from 31.2% to 39.9% (*p* < 0.001). However, those who reported performing intense physical activity 3-4 times/week prior to confinement decreased from 25 to 18.5% (*p* < 0.001). Moreover, Table 3 shows there was a statistically significant reduction in the PASBI score during confinement among all age groups, education levels, marginalisation, and places of residence.

Observing physical activity frequency in Table 3, the one performed most frequently among the population was 3 to 4 times per week before and during confinement, but even this category of weekly exercise frequency decreased by 6.3%, being more accentuated in men (6.5%) than in women. In women, weekly exercise was more affected in the category of 5 to 6 times and several times a day where a decrease occurred (−3.2 and −0.3%, respectively).

### 3.4. Lifestyle Quality Index (LSQI)

Some (29.2%) of the participants described having gained body weight, 6.1% stopped smoking, and 12.1% stopped consuming alcoholic beverages. However, 3% smoked more and 10.2% consumed more alcohol. With regard to sleeping habits, 53.3% of the participants reported sleeping later, 27% had interrupted night sleep, and 18% had disturbed sleeping schedules (*p* > 0.001). Nearly half of the participants (47.4%) were concerned about their health.

Table 4 describes the differences in LSQI by the participant’s characteristics. The majority of participants reported that during confinement they would leave their homes only if it was necessary (85.5%), 43% increased their time in front of screens, 81.6% spent more hours sitting or lying down, and 42.3% consumed food in front of screens once a day. The LSQI for the general population prior to confinement was 42 points (IQR 35–50) (m = 42.35, 95% CI 42.07–42.94) and 41 points during the confinement (IQR 34–47). As a possible consequence of the changes in the quality of diet and physical activity, the LSQI was also affected.

## 4. Discussion

Nutrition is one of multiple factors that determine the immune response [21]. Poor nutritional status and pre-existing noncommunicable diseases (obesity, diabetes mellitus, cardiovascular, and chronic lung diseases, among others) compromise the immune system. Diet and lifestyle are key elements that affect patient outcomes in severe infections, and it has been suggested that they may play a role in COVID-19 [22]. Therefore, diet quality and lifestyle may be key in mitigating the risks associated with COVID infection. Notably, there are several significant risk factors for severe COVID-19.

ESCAN-COVID19Mx aimed to evaluate the perceived changes in the quality of diet, lifestyle, emotional eating, and physical activity of Mexican adults with access to the Internet before and confined during the COVID-19 pandemic. A total of 8289 Mexicans who reported staying at home during the confinement period and had Internet access completed the online questionnaire.

The results from our study showed an improvement in diet quality measured through a DQI; among those, mainly women, who were in an age range of 18 to 59 years, had a normal weight or overweight, higher education level, lived in Mexico City, the north and centre of the country, and who lived in areas of low marginalisation.

These results were unexpected since, when registering the hoarding of frozen and canned foods at the beginning of the pandemic [1,4,5], we anticipated a possible increase in the consumption of ultra-processed foods resulting in a lower diet quality of the diet. However, a third of the respondents had greater access to homemade foods and greater consumption of fresh foods during the stage of confinement.

In parallel with our study, results from a survey by National Institute of Public Health in Mexico (ENSARS-CoV-2) showed that during the COVID-19 confinement, their participants had a higher consumption of cereals, fresh fruits and vegetables, as well as meat, fish and shellfish, eggs, dairy products, legumes, oils, condiments, coffee and tea. Also, they increased their consumption of sweetened beverages, sweets and snacks. Unlike our study, their report describes a diet diversity index with an average score of 13.5 (0–18) [9]. According to ENSANUT 2020 COVID-19 (Mexican National Health and Nutrition Survey 2020 COVID-19), 39.5% of households presented negative dietary changes (increased consumption of foods not recommended or decreased consumption of recommended food groups for daily consumption). Among the households that presented positive changes in food consumption, more than 40% of households reported a decrease in their consumption of sweetened beverages, pastries, snacks, and sweets, and 44% of households reported an increase in consumption of vegetables and fruits. Among the homes that presented negative changes in their diet, 66% reported a decrease in the consumption of meat and fish, and more than 50% reported a decrease in the consumption of vegetables and fruits. The prevalence of food insecurity was higher in households with negative changes compared with households unchanged and with positive changes [23].

In France, the results of the Nutri-Net cohort showed a reduction in the consumption of fresh foods and an increase in foods high in sugar at the beginning of the COVID-19 pandemic [24]. On the contrary, the COVIDiet study in Spain showed that during confinement, participants showed a greater adherence to the Mediterranean diet, which is characterized by the consumption of olive oil, fruits, and vegetables, and evidence suggests it is optimal to prevent non-communicable diseases and preserve good health [25].

Recent studies in other countries have also investigated eating behaviours during the COVID-19 pandemic, reaching conflicting results [24,25,26,27,28]. However, the way negative emotions impact food consumption and their possible impact on body weight has only been assessed in an adolescent population [29]. Using the AEBQ-Esp, we measured the participants’ emotional overeating and emotional undereating during the COVID-19 confinement. During the validation of the AEBQ tool in Mexicans prior to confinement, the mean of eating more through emotions was 2.54 (± 0.87) and eating less through emotions was 2.79 (SD: 0.88) [15]. The results from this study showed no associations with either emotional over or undereating and BMI. These results differ with other publications that show that emotional overeating is positively associated with BMI in adult populations [12,15,30].

The results from our study suggest that confinement negatively impacted on the physical activity of the population. Our results are similar to findings from studies in other countries [25,31], which reported a reduction in physical activity and an increase in sedentary time during COVID-19 confinement. This reduction in physical activity was expected due to the closure of gyms and recreation centres, as people do less active commuting. It could be possible that reductions in physical activity could have been due to an excess of activities while at home, such as looking after children, home schooling, plus individual home office work, affecting more women than men [32]. This could partly explain the results obtained in this study and why women reduced their exercise frequency. Reduction in physical activity during confinement is alarming, as prior to the pandemic, less than half (42%) of the adults in Mexico referred doing physical-sports practice in their free time [9]. This reduction could have a long-term impact on lifestyle and could lead to the appearance of type 2 diabetes and metabolic syndrome [18]. In this study, we found that the sedentary lifestyle increased from 31.3 to 40%, while in the ENSARS-CoV-2 survey, the sedentary lifestyle rose from 39.5 to 45% [9]. Again, and as seen with diet quality, the physical activity and sedentary lifestyle indices were different, regardless of the degree of marginalisation and the region of the country (*p* < 0.001).

The LSQI, constructed with information on smoking, alcohol consumption, diet quality, physical activity, and sedentary behaviour, showed a deterioration in lifestyle factors in both young adults and middle age adults during COVID-19 confinement, but not in the elderly. However, it is striking that the change was not significant in participants with BMI < 18 kg/m^2^ (*p* = 0.298) and ≥ 30 kg/m^2^ (*p* = 0.609); with a basic education level (*p* = 0.081); medium, low, to very low degree of marginalisation (*p* = 0.217, 0.855, and 0.825, respectively); and those who lived in the southern region of Mexico (*p* = 0.096). Studies prior to the COVID-19 pandemic showed that people in social isolation were likely to initiate or increase behaviours such as smoking or alcohol consumption [6]. Nevertheless, to our knowledge, none of the studies in the context of the COVID-19 pandemic have shown a significant increase in tobacco and alcohol consumption. A study in Spain reported a decrease in the consumption of alcoholic beverages by 57.3% [25], while in Italy a study showed a significant reduction in smoking [31].

In Mexico, the prevalence of excessive alcohol consumption during COVID-19 confinement was 40.43%, being higher in men than women (56.49 and 25.70%, respectively), in individuals aged 20 to 29 years, and those who were single and with more education. A higher prevalence was observed in those who had to go out to work daily during confinement and those who worked from home all the time. In women, high risk consumption of alcohol was higher in urban areas compared to rural areas. Both in men and women, the possibility of excessive alcohol consumption increased with a higher level of schooling and decreased with age [33]. This is in agreement with our findings where urban young women with a higher education and low level of marginalisation predominated.

In our study, some of the participants reported having changes in their smoking and alcohol habits, both positive and negative; however, the majority these behaviours remained constant as smoking increased by 3%, while alcohol consumption rose to 10.2%. This situation is in contrast with the findings of the ENSARS-CoV-2, where cigarette consumption increased to 21% and alcohol to 12% [9].

It is possible that the reduction of tobacco and alcohol consumption may have been for health concerns or that participants solely engaged in these behaviours socially. More research is needed to understand the factors that influenced these behaviour changes and whether these changes were permanent. Sleeping habits were affected in more than a quarter of the participants. Our results were similar to those in ENSARS-CoV-2 [9]. It is worth noting that in our study, 45% of the participants had a BMI above 25 kg/m^2^, while it is known that the prevalence of overweight and obesity in Mexico is around 75% [10]. This could be related to the socioeconomic level of the population in the study, where marginalisation was low and there was a higher level of education.

Confinement during the pandemic magnified existing gender inequalities in Mexico. Domestic violence against women is not uncommon and may have been exacerbated due to economic uncertainty, overcrowding at home, and limited access to support networks. In addition, during quarantine, many women were responsible for increased household chores, caring for children, helping with their school activities as all schools closed during the pandemic, and taking care of sick and elderly family members (including family members with COVID) [32]. This may have influenced time available for cooking healthy options and for engaging in physical activity.

### Limitations and Strengths

One possible limitation of ESCAN-COVID19Mx is that the instrument did not collect information on the type of economic activities, household income, or food security of the participants. In addition, we did not inquire on the reasons for making the behaviour changes and we did not reapply the survey at the end of confinement. Therefore, it is not possible to understand the causality of these changes or whether these were maintained. The recruitment strategy through social networks and snowball methodology resulted in a sample that consisted mainly of women, with an age range of 18 to 38 years, with a high level of education, and living in areas with very low levels of marginalisation, with access to the Internet, and therefore it is not representative of the Mexican population [34]. Due to the cross-sectional nature of our study, causal inferences should not be made between sociodemographic characteristics and changes in the quality of diet, physical activity, and lifestyle. Likewise, participants self-reported their weight and height; therefore, it is possible that their BMI could be under or overestimated.

Despite the study’s limitations, the main strengths of our study are that, to our knowledge, this is the first report with a sample size greater than the estimated that obtained responses from all over the country about the changes in diet quality, physical activity, and lifestyle in Mexicans who stayed home during COVID-19 confinement. The results from this study underscore the importance of improving diet quality in people who stay at home or work from home. A possible reason for this could be that most of the people who participated in this survey continued working from home during lockdown, however in this study we did not collect information about the participant’s current working status to confirm this.

ESCAN-COVID19Mx was carried out before the new Mexican frontal warning labelling regulation came into place on the 1 October 2020 [35]; thus, this survey could serve as a reference point to evaluate changes in food consumption for this new food ruling.

The COVID-19 pandemic has emphasised the importance of preventing chronic degenerative diseases, as evidence suggests that people living with these conditions are at higher risk of developing a more severe condition [36]. Therefore, it is important to encourage healthy lifestyle behaviours among the population during and after this time of crisis to prevent the risk of complications due to infectious and chronic degenerative diseases. One of the main challenges that are being faced during COVID-19 confinement is that there is a lack of resources and programs that help the general population improve their health under these conditions.

## 5. Conclusions

The results from this study showed that during COVID-19 confinement, Mexicans with access to the Internet that were staying at home and who had a low degree of marginalisation and a higher educational level, increased the quality of their diet through consuming more fresh foods prepared at home, yet they also increased the consumption of sugar-sweetened beverages, reduced their physical activity and possibly the general quality of their lifestyle decreased.

## Figures and Tables

**Figure 1 nutrients-13-04256-f001:**
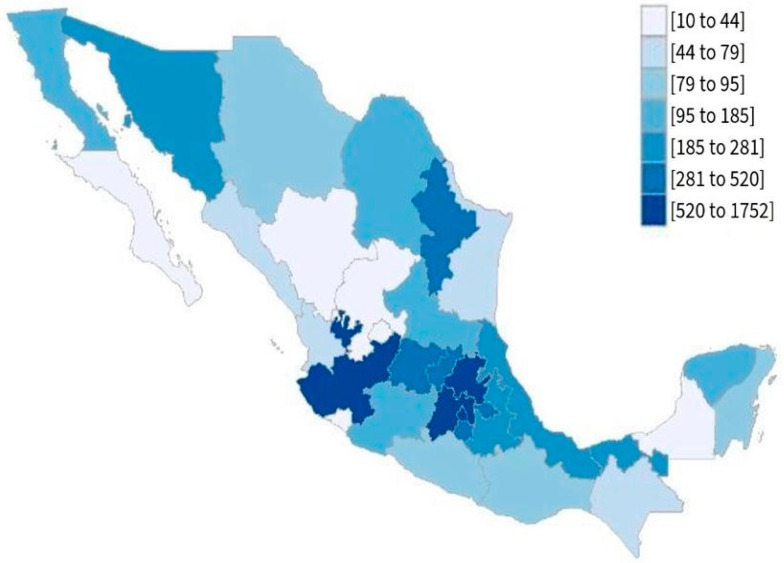
Distribution of ESCAN- COVID19Mx responses in Mexico.

**Figure 2 nutrients-13-04256-f002:**
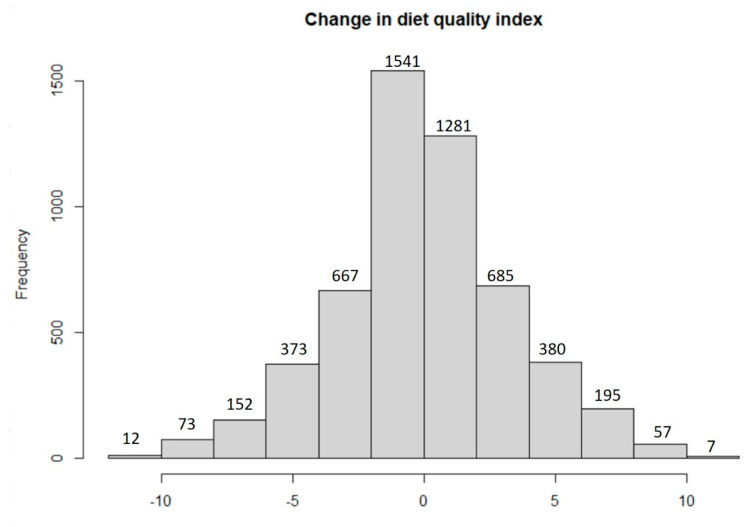
Change in the Mini-ECCA scale prior and during the confinement.

**Table 1 nutrients-13-04256-t001:** ESCAN-Covid19mx sample characteristics.

Characteristic		Total *n* = 8289	Female *n* = 6632	Male	*p*-Value
				*n* = 1657	
Sex			80%	20%	<0.001
Age, years (IQR)		34.6	34.3	35.5	0.001
		(34.4–34.8)	(34.1–34.6)	(34.9–36.1)	
	18–28	39.60%	39.70%	39.30%	0.793
	29–38	30.40%	31.30%	26.60%	<0.001
	39–48	18.00%	17.90%	18.50%	0.627
	49–58	8.90%	8.60%	10.30%	0.03
	59–79	3.10%	2.50%	5.30%	<0.001
Education					
	Basic	15.80%	15.40%	17.70%	0.023 **
	Higher	84.20%	84.60%	82.30%	0.023 **
Place of residence					
	Mexico City	21.90%	21.70%	22.70%	0.45
	Centre of Mexico	45.80%	45.90%	45.30%	0.633
	North of Mexico	14.90%	15.60%	12.20%	0.001
	South of Mexico	17.40%	16.80%	19.90%	0.005
Level of marginalisation					
	High	2.10%	2.00%	2.10%	0.999
	Medium	3.60%	3.60%	3.90%	0.932
	Low	11.50%	11.50%	11.60%	0.819
	Very low	82.80%	82.90%	82.40%	0.175
BMI, kg/m^2^ (IQR)		24.5	24.1	25.9	<0.001
		(22.1–27.5)	(21.8–27.1)	(23.9–28.7)	
	≤18	2.20%	2.50%	1.30%	0.004
	18.1–24.9	53.10%	57.10%	36.80%	<0.001
	25–29.9	31.40%	28.30%	44.00%	<0.001
	≥30	13.30%	12.00%	18.00%	<0.001

** proportion differences at 0.01 level (z-test). All the rest were treated with mean differences.

**Table 2 nutrients-13-04256-t002:** Characteristics of the participants according to the Mini-ECCA scale before and during the COVID-19 confinement.

Characteristic		*n*	Before the Confinement, m (IQR)	During the Confinement, m (IQR)	*p*-Value
Age, years					
	18–28	3282	8 (6–10)	8 (7–10)	<0.001
	29–38	2518	8 (6–9)	9 (7–10)	<0.001
	39–48	1494	8 (6–10)	9 (7–10)	<0.001
	49–58	740	8 (7–10)	9 (8–11)	<0.001
BMI kg/m^2^					
	≤18	185	7.5 (6–9)	8 (7–10)	0.011
	18.1–24.9	4380	8 (6–10)	9 (7–10)	<0.001
	25–29.9	2595	8 (6–10)	9 (7–10)	<0.001
	≥30	1096	8 (6–10)	9 (7–10)	<0.001
Education					
	Basic	1312	7 (5–9)	8 (7–10)	<0.001 **
	Higher	6977	8 (6–10)	9 (7–10)	<0.001 **
Level of marginalization					
	High	120	9 (6.5–10.5)	9 (7–11)	<0.001
	Medium	211	8 (6–10)	9 (7–10)	0.218
	Low	672	8 (6–10)	9 (7–10)	<0.001
	Very low	4832	8 (6–10)	9 (7–10)	0.277
Place of residence					
	Mexico City	1752	8 (6–10)	9 (7–10)	<0.001
	North of Mexico	1193	8 (6–10)	8 (7–10)	0.001
	Centre of Mexico	3663	8 (6–10)	9 (7–10)	<0.001
	South of Mexico	1390	8 (6–9)	9 (7–10)	0.001

m—median; IQR—interquartile range. ** proportion differences at 0.01 level (z-test). All the rest were treated with mean differences.

**Table 3 nutrients-13-04256-t003:** Characteristics of the participants according to the Physical Activity and Sedentary Behaviour index (PASBI), prior and during COVID-19 confinement.

Characteristic		*n*	Before Confinement, m (IQR)	During Confinement, m (IQR)	*p*-Value
Age, years					
	18–28	3282	9 (5–14)	6 (2–10)	<0.001
	29–38	2518	9 (5–13)	5 (2–10)	<0.001
	39–48	1494	9 (5–14)	7 (3–11)	<0.001
	49–58	740	9 (5–14)	7 (3–11)	<0.001
	59–79	254	9.5 (5–14)	8 (5–12)	<0.001
BMI kg/m^2^					
	≤18	185	8 (5–12)	5 (2–9)	<0.001
	18.1–24.9	4380	10 (5–14)	7 (3–11)	<0.001
	25–29.9	2595	9 (5–14)	5 (2–10)	<0.001
	≥30	1096	8 (4–11)	5 (2–9)	<0.001
Education					
	Basic	1312	9 (5–13)	5 (2–10)	<0.001 **
	Higher	6977	9 (5–14)	6 (2–11)	<0.001 **
Level of marginalisation					
	High	120	9 (5–14)	7 (3–11)	<0.001
	Medium	211	9 (5–13.5)	5 (3–9.5)	<0.001
	Low	672	9 (5–14)	7 (3–11)	<0.001
	Very low	4832	9 (5–14)	6 (3–11)	<0.001
Place of residence					
	Mexico City	1752	9 (5–13)	5 (2–10)	<0.001
	North of Mexico	1193	9 (5–14)	6 (2–10)	<0.001
	Centre of Mexico	3663	9 (5–14)	7 (3–11)	< 0.001
	South of Mexico	1390	9 (5–14)	5 (2–11)	< 0.001

m—median; IQR—interquartile range. ** proportion differences at 0.01 level (z-test). All the rest were treated with mean differences.

**Table 4 nutrients-13-04256-t004:** Characteristics of the participants according to the Lifestyle Quality Index (LSQI) prior and during COVID-19 confinement.

Characteristic		*n*	Before Confinement, m (IQR)	During Confinement, m (IQR)	*p*-Value
Age, years					
	18–26	3282	43 (35–50)	41 (35–47)	<0.001
	29–36	2518	42 (35–49)	40 (33–46)	<0.001
	39–46	1494	44 (37–51)	42 (35–47)	<0.001
	49–56	740	45 (36–52)	42 (37–48)	<0.001
	59–77	254	46 (40–52)	45 (39–51)	0.535
BMI kg/m^2^					
	≤18	185	42 (35–48)	40 (35–47)	0.298
	18.1- 24.9	4380	44 (37–51)	42 (35–48)	<0.001
	25- 29.9	2595	43 (34–50)	41 (34–46)	<0.001
	≥30	1096	44 (32–47)	40 (33–46)	0.609
Education					
	Basic	1312	43 (35–50)	42 (36–48)	0.081 **
	Higher	6977	43 (35–50)	41 (34–47)	<0.001 **
Level of marginalisation					
	High	120	42 (36–51)	42 (37–50)	<0.001
	Medium	211	43 (36–51)	43 (37–48)	0.217
	Low	672	44 (33–50)	43 (37–48)	0.855
	Very low	4832	43 (35–50)	41 (35–47)	0.825
Place of residence					
	Mexico City	1752	43 (35–49)	40 (33–46)	<0.001
	North of Mexico	1193	42 (35–49)	41 (34–47)	0.001
	Centre of Mexico	3663	43 (35–51)	41 (35–47)	<0.001
	South of Mexico	1390	43 (36–50)	42 (36–49)	0.096

m—median; IQR—interquartile range. ** proportion differences at 0.01 level (z-test). All the rest were treated with mean differences.

## Data Availability

Data are available from S.E.M.-V. upon reasonable request.

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
