# Peer review of "Perceived Diet Quality, Eating Behaviour, and Lifestyle Changes in a Mexican Population with Internet Access during Confinement for the COVID-19 Pandemic: ESCAN-COVID19Mx Survey"

_nutrients, 2021, doi:10.3390/nu13124256_

Round 1

Reviewer 1 Report

The manuscript proposed by Martínez-Vázquez and collaborators described the effect of COVID-19 disease in the Mexican population, this specific nutrition scientific field is so interesting and could monitor new trends in nutritional habits and consumers changes influenced by the pandemic situation. However, some major weaknesses limited the present article:

1.- The manuscript reported a good number of people interviewed, even so I suggest they reconsider the title and/or some parts of the introduction and discussion. They ranged 80% of women, 70% of 18-38 years old and 84.2% higher education, these data, are the representation of the Mexican population?

2.- Introduction is so lax and should be more focused on the Mexican situation.

3.- Material and methods needs to be improved, and the survey transcription should be included in a supplementary field.

4.- Result description is extremely confused caused by the pour description of MM. Results should be rephrased according to the questionary and described with details.

Minor

Why is included “)” next to the citation?

Some part of the text needs to be rephrased, for example, “Some (29.2%) of the participants” should be replaced by Some of the participants (29.2%).

The manuscript included DQI in some parts and in others IQR

Author Response

Response to Reviewer #1

The manuscript proposed by Martínez-Vázquez and collaborators described the effect of COVID-19 disease in the Mexican population, this specific nutrition scientific field is so interesting and could monitor new trends in nutritional habits and consumers changes influenced by the pandemic situation. However, some major weaknesses limited the present article:

Thank you very much for your helpful comments on our paper “Perceived diet and lifestyle changes during confinement for Covid-19 pandemic in Mexico: ESCAN-COVID19Mx Survey”. We have made the following changes regarding your observations, and made the suggested changes in the paper, as follows:

1.- The manuscript reported a good number of people interviewed, even so I suggest they reconsider the title and/or some parts of the introduction and discussion. They ranged 80% of women, 70% of 18-38 years old and 84.2% higher education, these data, are the representation of the Mexican population?

Thank you so much for your suggestions. We have taken into account your comments and thus have altered the title of the paper as follows:

“Perceived diet quality, eating behaviour and lifestyle changes in a Mexican population with Internet access during confinement for Covid-19 pandemic: ESCAN-COVID19Mx Survey”

We have therefore in turn added several sections to the results and discussion (including limitations):

Lines 330-333: Results from our study showed an improvement in diet quality measured through a DQI, among those, mainly women, who were in an age range of 18 to 59 years, had a normal weight or overweight, higher education level, lived in Mexico City, the north and centre of the country, and who lived in areas of low marginalisation.

Lines 379-381: It could be possible that reductions in physical activity could have been due to an ex-cess of activities whilst at home, such as looking after children, home schooling, plus individual home office work, affecting more women than men.

Lines 405-413: In Mexico, the prevalence of excessive alcohol consumption during Covid-19 con-finement was 40.43%, being higher in men than women (56.49 and 25.70%, respectively), in individuals aged 20 to 29 years, single and with more education. A higher prevalence was observed in those who had to go out to work daily during confinement and those who worked from home all the time. In women, high risk consumption of alcohol was higher in urban areas compared to rural areas. Both, in men and women, the possibility of excessive alcohol consumption is increase with a higher level of schooling and decreases with age. REF This is in agreement with our findings where urban young women, with higher education and low level of marginalisation predominated.

Lines 428-435: Confinement during the pandemic magnified existing gender inequalities in Mexico. Domestic violence against women is not uncommon and may be exacerbated due to economic uncertainty, overcrowding at home, and limited access to support networks. Also, during quarantine, many women were responsible for increased house-hold chores; caring for children, helping with their school activities since all schools closed during the pandemic, taking care of sick and elderly family members (including family members with Covid-19). This may have influenced time for cooking healthy options and for engaging in physical activity.

Limitations. Lines 443-447: The recruitment strategy through social networks and snowball methodology resulted in a sample that consisted mainly of women, age range of 18 to 38 years, with a high level of. education, and living in areas with very low levels of marginalisation, with access to Internet, and therefore it is not representative of the Mexican population.

2.- Introduction is so lax and should be more focused on the Mexican situation.

Thank you very much for your comment, we have amended the introduction as follows:

Lines 90-98: Results from a telephone survey conducted by the Mexican Institute of Public Health (ENSARS-CoV-2) in which data from 1000 Mexicans was collected between May 11th and 30th 2020, showed participants had less diversity in their diet, reduced their level of physical activity, and increased sedentary behaviours and sleeping hours. Furthermore, according to ENSANUT 2020 COVID-19 (Mexican National Health and Nutrition Survey 2020 COVID-19) 39.5% of households presented negative dietary changes during COVID-19 confinement. Among the households that presented negative changes in their diet, 66% reported a decrease in the consumption of meat and fish, and more than 50% reported a de-crease in the consumption of vegetables and fruits.

Mexico has a high prevalence of overweight and obesity (75.2% in adults) which places its population at high risk of severe COVID-19.

3.- Material and methods needs to be improved, and the survey transcription should be included in a supplementary field.

Once again, we thank you for your comments.

We have included a transcription of the questionnaire used in Spanish in the Supplementary information as a separate file.

We improved the Materials and Methods section, adding some details for clarity.

Line 142: self-reported weight and height

Line 150: A Diet Quality Index (DQI) was calculated. 

Lines 159-161: However, participants were asked if they thought they would have given the same answer prior to confinement.  

Lines 183-184: We were interested in measuring the impact of different lifestyle factors as a composite score that comprised multiple factors [19,20].

Lines 193-195: Information on six lifestyle factors (Diet quality, physical activity, sedentary behaviour, smoking, alcohol consumption and sleeping habits) were used to construct the Lifestyle Quality Index (LSQI). For diet quality response values were from 1 to 4, for physical activity, response values were from 0 to 4 and, for diet quality from 1 to 4.  Subsequently, these values were multiplied by 10 to weight the value of each factor equally.  

4.- Result description is extremely confused caused by the pour description of MM. Results should be rephrased according to the questionary and described with details.

Thank you for your comment. We have revised our results section and have made changes according to your suggestions, as follows:

Line 222: We renamed this section: Sociodemographic characteristics. We reordered some of the information as follows:

Lines 234-235 Half of the respondents (53.1%) had a BMI between 18 and 24.9 kg/m2, a third had a BMI between 25 and 29.9 kg/m2 (31.4%), and 13.3% had a BMI ≥ 30 kg/m2.

Lines 242-247:  The sources used by the participants to keep themselves informed about the COVID-19 pandemic were social media (78.8%), leaders and health professionals (73%) and health authorities, government and health institutions (78.8%). Regarding money spent on food, 38.9% of the participants perceived having a higher food expenditure than usual during confinement, but 47.8% reported having a lower family income. Nearly half of the households (44%) included minors.

Lines 278-283: Observing physical activity frequency in Table 3, the one performed most frequently among the population was 3 to 4 times per week before and during confinement, but even this category of weekly exercise frequency decreased by 6.3%, being more accentuated in men (6.5%) than in women. In women, weekly exercise was more affected in the category of 5 to 6 times and several times a day where a decrease occurred (-3.2 and - 0.3%, respectively).

Line 306: Table 4 describes the differences in LSQI by the participant’s characteristics.

Minor

Why is included “)” next to the citation?

We have corrected this and eliminated “)” next to the citation.

Some part of the text needs to be rephrased, for example, “Some (29.2%) of the participants” should be replaced by Some of the participants (29.2%).

We revised the entire manuscript and improved it for clarity.

The manuscript included DQI in some parts and in others IQR

Thank you for your comment. There is no contradiction as DQI refers to Diet Quality Index, whereas IQR is the interquartile range.

Reviewer 2 Report

This study aims to evaluate the perceived changes in lifestyles, quality of diet, and eating behaviors of Mexican adults (> 18 years) during the COVID-19 confinement, according to sociodemographic characteristics and nutritional status. Based on their results, the authors concluded that Mexicans staying at home during the COVID-19 confinement perceived positive changes in the quality of their diet, smoking, and alcohol consumption, but negative changes in the physical activity levels and sleep quality.

The introduction section should be improved. I suggest shortening it, erasing the unnecessary information (i.e. the first paragraph reported several a lot of well-known information). I suggest presenting the thematic, inserting several missed references such as DOI: 10.1016/j.neuroimage.2021.118311, DOI:10.3390/medicina56120640, DOI 10.3390/ijerph18041376. This paragraph should be transferred in the material and methods “Results from a telephone survey conducted by the Mexican Institute of Public Health (ENSARS-CoV-2) in which data from 1000 Mexicans was collected between May 11th and 30th 2020, showed participants had less diversity in their diet, reduced their level of physical activity, and increased sedentary behaviors and sleeping hours [14]). Therefore, it becomes relevant to document the effects of the pandemic on the population’s eating habits and lifestyle, since an individual’s nutritional status and poor diet quality may impact on comorbidities and subsequent complications”. Finally, the aims should be clearly written. Please, check it.

The Material and Methods section and the result section reported the methods and results of the present study.

The discussion section should be improved. First of all, I suggest inserting several considerations about the food intake in order to mitigate the risks related to COVID-19 infection. In this regard, I suggest reading and inserting these missed references: DOI: 10.3390/nu13030976; DOI: 10.3390/ijms21093104. Moreover, the authors missed to insert the study's limitations. Particularly, the authors missed analyzing that their study they interviewed 80% of females. Their data suffered from this bias: the improvement in the food quality could be related to this data. Moreover, most participants were under 40 (70%): the benefit of the severe countermeasures to COVID-19 seems minimal in this age range compared to the old subjects. So their opinion is influenced by this situation. These important biases should be discussed and inserted in the limitation of the study.

Author Response

Response to Reviewer #2

This study aims to evaluate the perceived changes in lifestyles, quality of diet, and eating behaviors of Mexican adults (> 18 years) during the COVID-19 confinement, according to sociodemographic characteristics and nutritional status. Based on their results, the authors concluded that Mexicans staying at home during the COVID-19 confinement perceived positive changes in the quality of their diet, smoking, and alcohol consumption, but negative changes in the physical activity levels and sleep quality.

Thank you very much for your helpful comments on our paper “Perceived diet and lifestyle changes during confinement for Covid-19 pandemic in Mexico: ESCAN-COVID19Mx Survey”. We have made the following changes regarding your observations, and made the suggested changes in the paper.

The introduction section should be improved. I suggest shortening it, erasing the unnecessary information (i.e. the first paragraph reported several a lot of well-known information). I suggest presenting the thematic, inserting several missed references such as DOI: 10.1016/j.neuroimage.2021.118311, DOI:10.3390/medicina56120640, DOI 10.3390/ijerph18041376.

As you suggested, we have eliminated the first paragraph. We also added information pertaining the Mexican situation for context, as follows:

Lines 73-74: Studies on consumer behaviours describe that purchasing habits change in external contexts such as natural disasters or pandemics.

Lines 90-98: Results from a telephone survey conducted by the Mexican Institute of Public Health (ENSARS-CoV-2) in which data from 1000 Mexicans was collected between May 11th and 30th 2020, showed participants had less diversity in their diet, reduced their level of physical activity, and increased sedentary behaviours and sleeping hours. Furthermore, according to ENSANUT 2020 COVID-19 (Mexican National Health and Nutrition Survey 2020 COVID-19) 39.5% of households presented negative dietary changes during COVID-19 confinement. Among the households that presented negative changes in their diet, 66% reported a decrease in the consumption of meat and fish, and more than 50% reported a de-crease in the consumption of vegetables and fruits.

Line 99: Mexico has a high prevalence of overweight and obesity (75.2% in adults) which places its population at high risk of severe COVID-19.

This paragraph should be transferred in the material and methods “Results from a telephone survey conducted by the Mexican Institute of Public Health (ENSARS-CoV-2) in which data from 1000 Mexicans was collected between May 11th and 30th 2020, showed participants had less diversity in their diet, reduced their level of physical activity, and increased sedentary behaviors and sleeping hours [14]). Therefore, it becomes relevant to document the effects of the pandemic on the population’s eating habits and lifestyle, since an individual’s nutritional status and poor diet quality may impact on comorbidities and subsequent complications”.

This paragraph is important because it depicts the results of a telephone survey carried out by the National Institute of Public Health (Mexico). We left it in the Introduction section and did not transfer it to the Material and Methods section since it is not part of our study. Additionally, we provided information on the prevalence of overweight and obesity in Mexico and a study that linked this situation to severe Covid-10 in the Mexican population.

Lines 90-98: Results from a telephone survey conducted by the Mexican Institute of Public Health (ENSARS-CoV-2) in which data from 1000 Mexicans was collected between May 11th and 30th 2020, showed participants had less diversity in their diet, reduced their level of physical activity, and increased sedentary behaviours and sleeping hours. Furthermore, according to ENSANUT 2020 COVID-19 (Mexican National Health and Nutrition Survey 2020 COVID-19) 39.5% of households presented negative dietary changes during COVID-19 confinement. Among the households that presented negative changes in their diet, 66% reported a decrease in the consumption of meat and fish, and more than 50% reported a de-crease in the consumption of vegetables and fruits.

Lines 99: Mexico has a high prevalence of overweight and obesity (75.2% in adults) which places its population at high risk of severe COVID-19.

Finally, the aims should be clearly written. Please, check it.

We rephrased the aim of our study as follows: Thus, the aim of this study is to describe the perceived changes in the quality of diet, eating behaviour, physical activity and lifestyle of a group of Mexican adults with internet access who were confined during the Covid-19 pandemic.

The Material and Methods section and the result section reported the methods and results of the present study.

The discussion section should be improved. First of all, I suggest inserting several considerations about the food intake in order to mitigate the risks related to COVID-19 infection. In this regard, I suggest reading and inserting these missed references: DOI: 10.3390/nu13030976; DOI: 10.3390/ijms21093104.

Thank you for your excellent suggestion. We added:

Line 318-324: Nutrition is one of multiple factors that determine the immune response. Poor nutritional status and pre-existing noncommunicable diseases (obesity, diabetes mellitus, cardiovascular and chronic lung diseases, among others), compromise the immune system. Diet and lifestyle are key elements that affect patient outcomes in severe infections, and it has been suggested that they may play a role in COVID-19.  Therefore, diet quality and lifestyle may be key in mitigating the risks associated with Sars-Cov2 infection. Notably, there are several significant risk factors for severe COVID-19.

References: doi.org/10.1038/s41430-021-00949-8 and doi:10.3390/nu12051466

Moreover, the authors missed to insert the study's limitations. Particularly, the authors missed analyzing that their study they interviewed 80% of females. Their data suffered from this bias: the improvement in the food quality could be related to this data.

Moreover, most participants were under 40 (70%): the benefit of the severe countermeasures to COVID-19 seems minimal in this age range compared to the old subjects. So their opinion is influenced by this situation. These important biases should be discussed and inserted in the limitation of the study.

The study’s limitations are described at the end of the Discussion section. We have identified it properly via a subtitle (Limitations and strengths). We have added among the limitations the large proportion of women and young subjects in our sample.

Lines 438-451:  One possible limitation of the ESCAN-COVID19Mx is that the instrument did not collect information on the type of economic activities, household income or food security of the participants. Also, we did not inquire on the reasons for making the behaviour changes and we did not reapply the survey at the end of confinement. Therefore, it is not possible to understand the causality of these changes or whether these were maintained. The recruitment strategy through social networks and snowball methodology resulted in a sample that consisted mainly of women, age range of 18 to 38 years, with a high level of. education, and living in areas with very low levels of marginalisation, with access to Internet, and therefore it is not representative of the Mexican population. Due to the cross-sectional nature of our study, causal inferences should not be made between sociodemographic characteristics and changes in the quality of diet, physical activity, and lifestyle. Likewise, participants self-reported their weight and height; therefore, it is possible that their BMI could be under or overestimated.

Round 2

Reviewer 1 Report

I thank the authors for considering my suggestions.

Reviewer 2 Report

The authors have improved their manuscript sufficiently.